# Comparison of Different Missing-Imputation Methods for MAIAC (Multiangle Implementation of Atmospheric Correction) AOD in Estimating Daily PM₂.₅ Levels

**Zhao-Yue Chen** [1,2] 🟢, **Jie-Qi Jin** [2], **Rong Zhang** [3], **Tian-Hao Zhang** [4], **Jin-Jian Chen** [1], **Jun Yang** [5] 🟢, **Chun-Quan Ou** [2,*] 🟢 **and Yuming Guo** [1]

1. Department of Epidemiology and Preventive Medicine, School of Public Health and Preventive Medicine, Melbourne, VIC 3004, Australia; zhaoyue21@i.smu.edu.cn (Z.-Y.C.); cjj19960824@i.smu.edu.cn (J.-J.C.); yuming.guo@monash.edu (Y.G.)
2. State Key Laboratory of Organ Failure Research, Department of Biostatistics, Guangdong Provincial Key Laboratory of Tropical Disease Research, School of Public Health, Southern Medical University, Guangzhou 510515, China; jjq123@i.smu.edu.cn
3. Department of Urban Planning and Design, The University of Hong Kong, Pokfulam, Hong Kong; rong.zhang@connect.hku.hk
4. State Key Laboratory of Information Engineering in Surveying, Mapping and Remote Sensing, Wuhan University, Wuhan 430079, China; tianhaozhang@whu.edu.cn
5. Institute for Environmental and Climate Research, Jinan University, Guangzhou 511443, China; yangjun_eci@jnu.edu.cn

* Correspondence: ocq@smu.edu.cn; Tel.: +8620-61360456; Fax: +8620-61648319

**Abstract:** The immense problem of missing satellite aerosol retrievals (Aerosol Optical Depth, (AOD)) detrimentally affects the prediction ability of ground-level PM₂.₅ concentrations and may lead to unavoidable biases. An appropriate missing-imputation method has not been well developed to date. This study developed a two-stage approach (AOD-imputation stage and PM₂.₅-prediction stage) to predict short-term PM₂.₅ exposure in mainland China from 2013–2018. At the AOD-imputation stage, geostatistical methods and machine learning (ML) algorithms were examined to interpolate 1 km satellite aerosol retrievals. At the PM₂.₅-prediction stage, the daily levels of PM₂.₅ were predicted at a resolution of 1 km, based on interpolated AOD and meteorological data. The statistical performances of the different interpolation methods were comprehensively compared at each stage. The original coverage of retrieved AOD was 15.46% on average. For the AOD-imputation stage, ML methods produced a higher coverage (98.64%) of AOD than geostatistical methods (21.43–87.31%). Among ML algorithms, random forest (RF) or extreme gradient boosted (XG-interpolated) AOD produced better interpolated quality (CV $R^2$ = 0.89 and 0.85) than other algorithms (0.49–0.78), but XGBoost required only 15% of the computing time of RF. For the PM₂.₅ predicted stage, neither RF-AOD nor XG-AOD could guarantee higher accuracy in PM₂.₅ estimations (CV $R^2$ = 0.88 (RF or XG-AOD) compared to 0.85 (original)), or more stable spatial and temporal extrapolation (spatial, (temporal) CV $R^2$ = 0.83 (0.83), 0.82 (0.82), and 0.65 (0.61) for RF, XG, and original). For the AOD-imputation stage, the missing-filled efficiency depended more on external information, while the missing-filled accuracy relied more on model structure. For the PM₂.₅ predicted stage, efficient AOD interpolation (or the ability to eliminate the missing data) was a precondition for the stable spatial and temporal extrapolation, while the quality of interpolated AOD showed less significant improvements. It was found that XG-AOD is a better choice to estimate daily PM₂.₅ exposure in health assessments.

**Keywords:** machine learning; aerosol optical depth; missing replacement; short-term; PM₂.₅



## 1. Introduction

Since 1998, in China and India, industrialization, economic development, and a substantially increasing energy demand has led to over five times and triple growth in China's and India's coal-fired power, respectively [1]. Although accounting for 36% of the world's population, they have accounted for 59% of global disease burden attributable to fine-particulate pollution [2]. Many previous epidemiological studies around the world have linked short-term $PM_{2.5}$ exposure to emergency hospital admissions and even deaths from acute or chronic illnesses such as asthma and stoke [3–7]. Despite such severe air pollution conditions and widely publicized concerns, there are sparse PM monitoring stations, and this is a significant hurdle for assessing pollution exposure. Among the 7.5 billion people worldwide, only approximately 5.9 billion (approximately 80%) people live in regions that are covered by available PM readings [8]. This is because of the high cost of building and maintaining monitoring sites.

In recent years, there has been an increasing trend to use satellite aerosol optical depth (AOD) for estimating ground-level $PM_{2.5}$ concentrations. Compared with conventional monitoring, the lower cost, higher coverage, and higher spatial resolution are advantageous, but the inherent drawbacks of AOD are obvious and difficult to solve. The most critical issue is the high missing rate of satellite-retrieved AOD, potentially related to orbit patterns, cloudiness, polar night, and surface reflectivity [9,10]. Kahn et al. [10] reported that the fairly low likelihood (approximately 15%) of successfully extracting aerosol retrievals from satellite instruments is a global problem, and the high missing rate of aerosol retrievals makes it impossible to estimate short-term $PM_{2.5}$ exposure in continuous time series or most time in the research period [11–13]. Furthermore, some reported daily $PM_{2.5}$ measurements have presented different distributions between AOD-missing days and other days [14]. Compared with long-term pollution exposure, short-term health assessments rely more on integral temporal variations in exposure. A drawback of satellite-based estimates is that they cannot represent the real distribution of $PM_{2.5}$.

For addressing this issue, AOD retrievals need to be imputed before estimating $PM_{2.5}$ levels where AOD is an important input variable. Multiple geostatistical approaches have attempted to impute AOD. Some examples are inverse distance weighting, nearest neighbors, kriging, generalized additive models or multiple imputation [15–17], but have achieved relatively low efficiency (missing rate after interpolation approximately 30–50%) and quality (cross-validation R-square approximately 0.34–0.64). We previously proposed [8] a two-step (TS) interpolation (first using data from Terra to estimate Aqua missing values, then inverse distance weighted interpolation (IDW) for second step) to reduce the AOD-missing rates from 87.91% to 13.83%, and maintain a relatively satisfactory performance (CV R-square = 0.76). However, the coverage needs to be increased further, and higher efficiency of interpolations were also required, especially when the sample size increased with a higher resolution in (Multiangle Implementation of Atmospheric Correction) MAIAC AOD ($3^2$ times with resolution from 3 km to 1 km). Moreover, our previous method [9] was over-dependent on geostatistical interpolation and ignored some influences of other external information such as meteorological data and cloud fraction on the temporal dimension, which inevitably weakened the efficiency and accuracy of imputation. This study endeavors to improve upon some weaknesses in the previous study. Machine learning methods can assist with solving these problems due to their strengths in capturing complex non-linear relationships and high dimension interactions [9,18]. We attempted to select an optimal method by comprehensively comparing a variety of machine learning algorithms and geostatistical approaches in imputing missing data of AOD.

After imputing missing data of AOD, we previously confirmed that the combined method of non-linear exposure-lag-response model (NELRM) and XGBoost is superior to other nine ML models at a large spatial scale (CV $R^2$ = 0.86 vs. 0.54~0.83) [9], but it is still unknown how different AOD-imputation methods will affect the accuracy of $PM_{2.5}$ predictions.

Using a 6-year MAIAC AOD with a resolution of 1 km, this study aimed to compare the statistical performance of different geostatistical and machine learning (ML) algorithms at the AOD-imputation stage and also assess the influence of different imputation methods on the PM$_{2.5}$-prediction stage.

## 2. Materials and Methods

### *2.1. Materials*

#### 2.1.1. Satellite-Retrieved Product

A 1 km aerosol product covering China from 2013–2018 was acquired from the NASA Multiangle Implementation of Atmospheric Correction (MAIAC) AOD (https://ladsweb.modaps.eosdis.nasa.gov/) [19,20]. Compared with traditional retrieving algorithms, MAIAC can retrieve finer resolution aerosol data from the Moderate Resolution Imaging Spectro radiometer (MODIS) Collection 6 (C6) [20–22], and can improve AOD correction during cloud and snow, which is the vulnerable detection moment of a remote sensor [20]. Only the extracted AOD with a quality assurance flag will be used [21–23]. In addition, a 16-day 1 km Enhanced Vegetation Index (EVI) product and MODIS daily 5-km cloud fraction data (MOD06_L2) were obtained from the NASA website.

#### 2.1.2. Daily Monitoring Data

Daily site-level PM$_{2.5}$ measurements, collected by the tapered-element oscillating microbalance method (China MEE, 2016) in mainland China during 2014–2018 were obtained from the China National Environmental Monitoring Center. The start time of the study period was 2014 because the national PM$_{2.5}$ monitoring system was built in 2014 [9,24,25]. A total of 1605 monitoring sites in 385 cities are shown in Figure S1A.

#### 2.1.3. Meteorological Data and Land-Cover Data

Meteorological data, including the daily mean temperature, pressure, sunshine hours, water vapor pressure, precipitation, relative humidity, wind speed, and wind direction, were obtained from 839 meteorological stations (Figure S1B) during 2013–2018. For further integration, the Universal Kriging (UK) technique was employed to interpolate the daily site-level meteorological data into grid cells with a resolution of 1 km. The 6-hour planetary boundary layer height (PBLH) from the National Centers for Environmental Prediction (NECP) was daily averaged and resampled from 1 degree to 1 km grid cells [9].

### *2.2. Methodology*

The general workflow for estimating daily PM$_{2.5}$ exposure (including AOD-imputation stage and PM$_{2.5}$-prediction stage) is demonstrated in Figure 1.

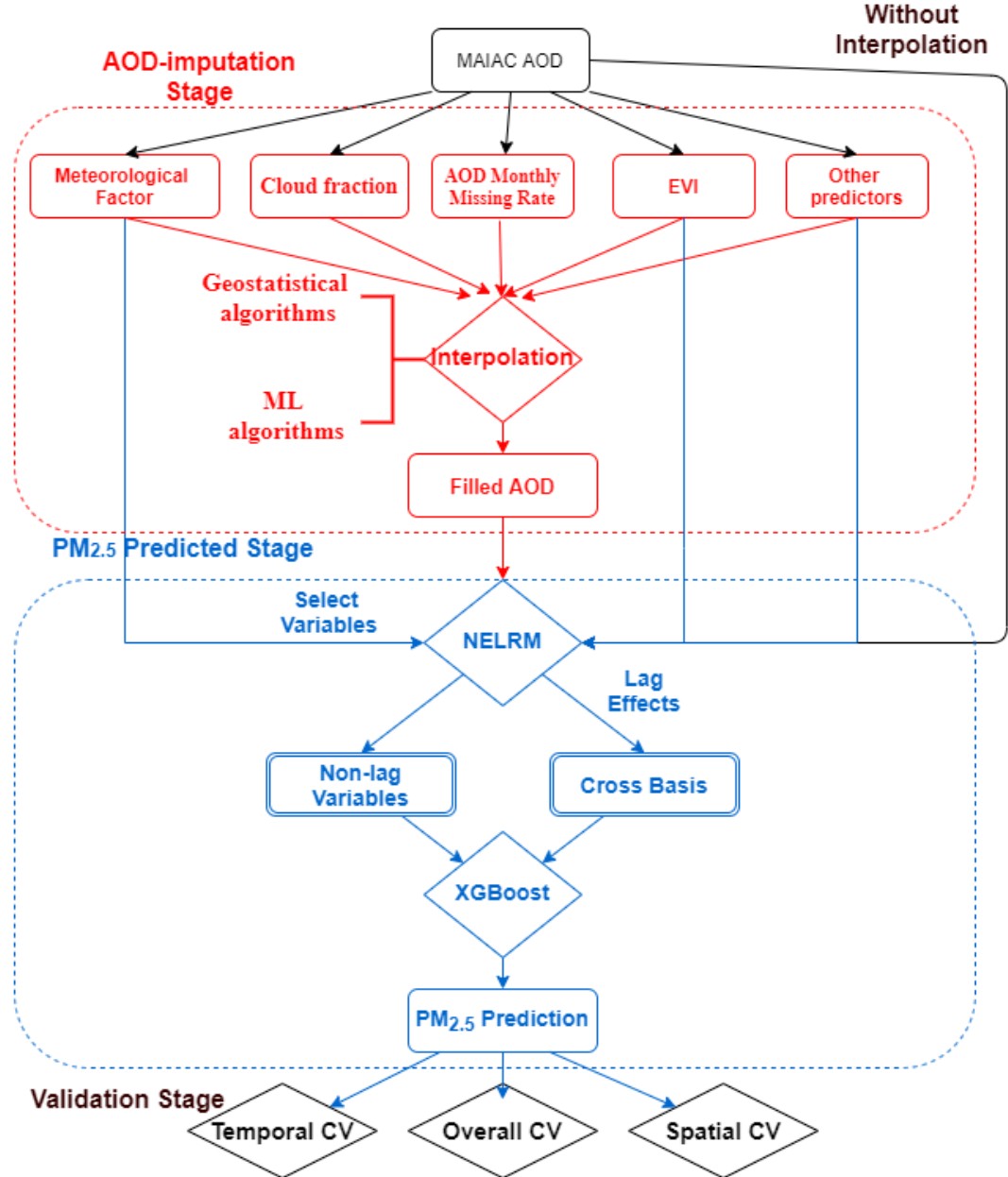

**Figure 1.** Framework for short-term PM$_{2.5}$ exposure estimation.

#### 2.2.1. AOD-Imputation Stage for MAIAC AOD

To select the optimal missing-filled methods for MAIAC AOD, we compared all potential methods, if the data required were available. These included different geostatistical interpolations [26–28], two-step imputation previously proposed [9] and other ML interpolating methods, including random forest (RF) [29], XGBoost (extreme gradient boosting) [30], support vector machine (SVM) [31], gradient boost model (GBM) [32], generalized additive model (GAM) [33], Bayesian regularized neural network (BRNN) [34], and least absolute shrinkage and selection operator (LASSO) [35]. The interpolation methods are summarized in Table 1. In general, geostatistical algorithms mainly depend on using spatial and temporal information in existing AOD data, while ML algorithms mainly use the complex relationship between AOD and other external information.

**Table 1.** Summary table of different interpolation methods at the AOD-imputation stage.

| Methods | Main Features | Merits | Demerits |
|---|---|---|---|
| **Geostatistical Algorithms** | | | |
| **TS** [9] | step I: mixed effect model<br>Step II: IDW | using different satellite products | more steps |
| **ST Kriging** [28] | regionalize spatio-temporal variable to obtain semi-variance for kriging | maximize use spatio-temporal neighborhood information | extensive computation for large spatial and temporal scale |
| **IDW** [28] | inversed distance weighted-average within search radius | easier and quicker to calculated | less accurate for complex spatial distribution |
| **Kriging** [28] | using semi-variance to explore spatial distribution within search radius | commonly used | not considering temporal variation |
| **NN** [28] | nearest neighbors within search radius | easiest and quickest to calculate | less spatial variation |
| **Machine Learning Algorithms** | | | |
| **LASSO** [35] | L1 regularization term was used to penalize nonessential or correlated features in model | prevent over fitting and ensure generalization | difficult to capture complex non-linear or interacting relationship |
| **GAM** [33] | use smooth functions to describe the relationship | fit non-linear relationship | easier to over fit |
| **GBM** [32] | an ensemble of weak prediction models, such as decision trees. | optimized by negative gradient of loss function. | over fitting and more computation time |
| **BRNN** [34] | Bayesian Inference was used to regularize the Maximum Likelihood in Neural Nets (similar regularization in Ridge Regression) | more robust than standard back-propagation nets | higher computation than L1 regularization |
| **SVM** [31] | Using kernel to map data into higher dimension, and then to fit the error within a certain threshold | produces higher accuracy with less computation power | weaker extrapolating ability in data with more noise |
| **RF** [29] | a meta estimator with numbers of classifying decision trees based on different sub-samples | control the overfitting by sub-samples | more computation time |
| **XG** [30] | a weighted ensemble of weak prediction models with regularized boosting and parallel processing | regularized boosting and parallel processing | less stable results in early stopping |

TS (two-step interpolation), ST Kriging (spatio-temporal kriging with 50 km buffer), IDW (inverse distance weighting with 50 km buffer), NN (nearest neighbors with 30 km buffer), LASSO (Least absolute shrinkage and selection operator), GAM (generalized additive model), GBM (gradient boost model), BRNN (Bayesian regularized neural network), SVM (support vector machine), RF (conditional inference random forest), and XG (extreme gradient boosting).

Geostatistical algorithms require less information and can suit the situation without requiring external information. In most cases, ML algorithms can provide a model for an entire study period, but geostatistical algorithms need to be conducted separately for each day or at specified intervals because they cannot provide a fixed model for the entire period. The parameter setting used in this study (Table S1) was tuning by 10-fold cross-validation.

For the ML algorithms, the selected predictors among different ML methods remained consistent to compare their performances objectively. All external information was selected by linear models with statistical significance ($P < 0.05$) and low variance inflation (VIF < 5, checks for multi-collinearity). The external information included PBLH, maximum ground surface temperature (Maxgst), precipitation (rain), maximum atmospheric pressure (Maxpres), mean relative humidity (MeanRH), sunshine duration (sunshine), mean ambient temperature (Meantemp), max/maximum wind speed (WS/MWS), cloud fraction (representing the levels of cloud coverage) from MOD06_L2, AOD monthly missing rate, EVI, and indicator variables such as longitude and latitude, altitude, day of the year, month, year, and day of the week. It should be noted that the sample size of MAIAC AOD covering China from 2013–2018 was too large (approximately 96,534,23 (numbers of grids) × 365 (days) × 6 (years)) to train the missing-filled model, and therefore, 5000 grids were randomly selected for each day as the modeling data (total sample size = 5000 (numbers of grids) × 365 (days) × 6 (years)). The external information for each grid was extracted from the corresponding grid cell.

### 2.2.2. PM$_{2.5}$-Prediction Stage

Due to natural geographical differences among the seven geographical regions in China such as climate, terrain, and vegetation [9], a satellite-PM$_{2.5}$ model was constructed separately for each region. To alleviate the discontinued and less-certainty problem between the regions' boundaries, the sites in the neighboring province (Figure S2) were also included in the corresponding modeling region.

Our previous PM$_{2.5}$ prediction model (a combined method of NELRM and XGBoost) [9] worked well at a large spatial scale (CV R$^2$ = 0.86) compared with other ML models (including random forest), and therefore, this study retained a similar PM$_{2.5}$ model structure. Here, the impacts of

different interpolations at the AOD-imputation stage on $PM_{2.5}$ predictive accuracy are mainly explored, rather than improving $PM_{2.5}$ model structure. The steps of the NELRM-XGBoost model have been previously described [9] and will be introduced briefly. First, non-lagged variables $X_1$ such as land-related data, and potential lagged variables $X_2$ such as original AOD, and meteorological variables were selected in the NELRM [14], with statistical significance (P < 0.05) and low variance inflation (VIF<5, checks for multi-collinearity). The optimal predictor combinations in different regions are shown in Table S2). Additionally, a cross-basis $Cb.X_2$ was constructed, explaining the lag effects, with degree of freedom (df = 3) for natural cubic smooth function and maximum 0–1 lagged days according to the 10-fold CV results. Finally, the weak learner $f_k(X_1, Cb.X_2)$ in XGBoost learning process was built, which was optimized by the loss function $L^k(\theta_k)$, adjusted by a previous iteration, and the regularizing term $\Omega^k(\theta_k)$, and hence, reducing model complexity for avoiding over fit. This is presented as the following equation:

$$\hat{y}_i^{(t)} = \sum_{k=1}^{t} \varepsilon \gamma_k f_k(X_1, Cb.X_2) \tag{1}$$

where $\hat{Y}_i^{(t)}$ denotes $PM_{2.5}$ estimation in iteration t; $\gamma_k$ and $\varepsilon$ are the weight vector and the learning rate. The setting in the satellite-$PM_{2.5}$ model (Table S1) was optimized by maximizing the 10-fold CV $R^2$ of the XGBoost approach for estimating $PM_{2.5}$.

### 2.2.3. Validation Stage

To better validate the performance of two stages, three kinds of 10-fold CV [14], were deployed, including overall CV, spatial CV, and temporal CV. All 10-fold CV were repeated 20 times. The main difference among them was randomly separating the dataset by observations, the location of sites, and date.

The overall CV is a common measurement that represents the performance stability in modeling dataset [13,23,36–38].

Spatial CV or spatial extrapolation is a more important indicator of prediction model of $PM_{2.5}$, because most predicted locations for exposure assessment do not have any observation due to the limited numbers of sites (1605). For a better description of the spatial performance distribution, we also conducted a Leave-One-Out-Cross-Validation ($LOOCV_{site}$), which leaves one site for validation each time. The $LOOCV_{site}$ results were further interpolated by UK into 1 km grids in China (interpolating quality: CV $R^2$ = 0.86), which simply represents spatial extrapolation in different predicted locations.

The temporal CV, or temporal extrapolation means the performance stability in different time points, as the training and validating datasets randomly split by dates in each fold. Furthermore, the validation analysis was performed in different months or years to examine the performance stability (Figure S3).

## 3. Results

### 3.1. Descriptive Statistics of MAIAC AOD and $PM_{2.5}$ Concentration

From 2013 to 2018, the daily coverage of MAIAC AOD in China generally remained at an average of approximately 15–16% (Figure 2). The highest coverage day was on 9 October 2013, reaching 33.22%, and the lowest coverage day was 4.55% on 3 July 2018. June–July and January–February accounted for 42.24% and 28.87% of the lower-coverage (≤10%) days, respectively, and more than 62.31% of the higher-coverage (≥20%) days were from September–December. The median value of the observed MAIAC AOD during the study period was approximately 0.31(interquartile range (IQR): 0.34).

From 2014 to 2018, the median $PM_{2.5}$ concentrations reported by the monitoring sites gradually decreased from 45.60 μg/m$^3$(IQR = 43.89) to 32.14 μg/m$^3$ (IQR = 30.34). The $PM_{2.5}$ concentrations in China were always higher in the cold season (October–March).

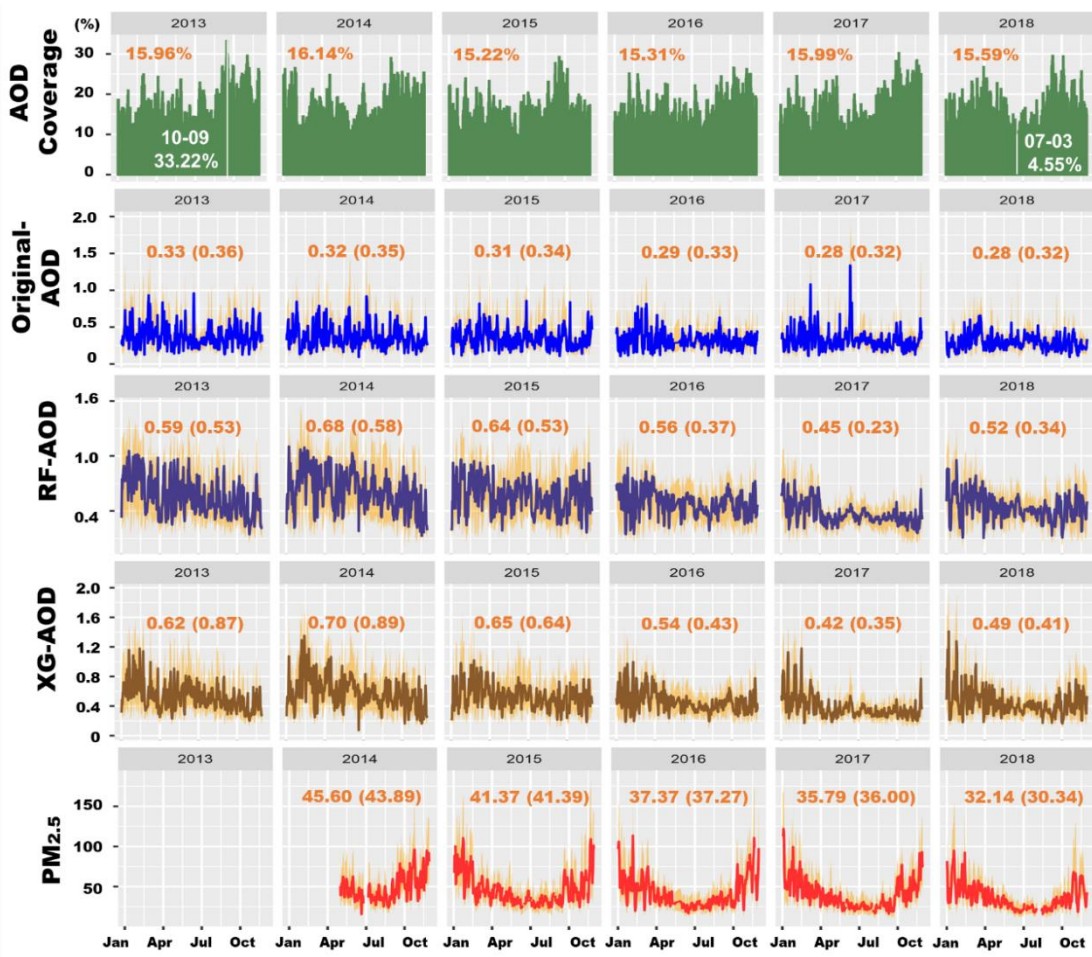

**Figure 2.** Descriptive statistics (coverage (%) and median (IQR)) of MAIAC AOD product, interpolated AOD, and observed ground-level PM$_{2.5}$ concentrations.

### 3.2. AOD-Imputation Stage

Compared with the geostatistical interpolation methods (coverage after interpolation approximately 21.43–87.31% for NN, Kriging, IDW, ST Kriging, and TS) (Table 2), the ML methods, using the same external information, were more efficient in filling the gaps, with almost full coverage (approximately 98.64%). The remaining missing data could not be fully eliminated due to the missing rates of external information. Among these algorithms, the RF and XG methods outperformed other algorithms (CV R$^2$ = 0.89, 0.85, and 0.49–0.78 for RF, XG, and other algorithms, respectively). Although the RF method was the most accurate, its computation time was longer than other ML algorithms and most geostatistical algorithms. The XG method was less time-consuming, had better performance, and higher coverage than our previous TS methods, although the accuracy was slightly lower than RF. Similar results were obtained for spatial CV and temporal CV (Table S3). Among the interpolations with higher accuracy (CV R$^2$ ≥ 0.7), we found the median AOD level always tended to increase (from 0.31 to 0.44–0.65) after geostatistical or ML interpolation. Owing to the same coverage in ML-interpolated AOD product (same external information for all ML algorithms), they had a more similar interpolated AOD distribution than the geostatistical interpolation. In addition, the linear correlation between interpolated AOD and PM$_{2.5}$ weakened after the interpolation (Table S4).

**Table 2.** Comparison of geostatistical and machine learning (ML) interpolation for MAIAC AOD product.

| Interpolation Method | Coverage (%) | Computation Time [a] | CV RMSE | CV $R^2$ | CV MAPE (%) |
|:---:|:---:|:---:|:---:|:---:|:---:|
| **Before Interpolation** | 15.46 | | | \ | |
| *Geostatistical Algorithms* | | | | | |
| **TS** | 87.31 | 55:38:56.56 | 0.17 | 0.75 | 20.56 |
| **ST Kriging** | 67.73 | 128:56:43.56 | 0.17 | 0.78 | 20.36 |
| **IDW** | 45.22 | 45:35:28.39 | 0.18 | 0.65 | 21.35 |
| **Kriging** | 42.37 | 88:46:37.24 | 0.17 | 0.66 | 20.89 |
| **NN** | 21.43 | 15:39:27.93 | 0.19 | 0.49 | 25.38 |
| *ML Algorithms* | | | | | |
| **RF** | 98.64 | 120:55:28.65 | 0.15 | 0.89 | 18.00 |
| **XG** | 98.64 | 18:00:38.20 | 0.15 | 0.85 | 19.06 |
| **SVM** | 98.64 | 19:04:47.64 | 0.17 | 0.72 | 19.41 |
| **BRNN** | 98.64 | 18:45:36.22 | 0.17 | 0.70 | 22.39 |
| **GBM** | 98.64 | 06:35:47.65 | 0.18 | 0.69 | 25.17 |
| **GAM** | 98.64 | 01:05:38.20 | 0.17 | 0.62 | 21.88 |
| **LASSO** | 98.64 | 01:18:30.23 | 0.19 | 0.49 | 28.03 |

TS (two-step interpolation), IDW (inverse distance weighting), ST Kriging (spatio-temporal kriging), NN (nearest neighbors with 30 km buffer), RF (Conditional Inference Random Forest), XG (extreme gradient boosting, XGBoost), SVM (support vector machine), BRNN (Bayesian regularized neural network), GBM (gradient boost model), GAM (generalized additive model), and LASSO (least absolute shrinkage and selection operator); [a] tested by Windows 10 system in 3.4 GHz computer with 16 GB of RAM.

### 3.3. PM$_{2.5}$ Predicted Stage

Six AOD-interpolation methods allowed relatively higher performances (with CV $R^2$ after over 0.7), and therefore, their interpolated AOD product was used to build the daily PM$_{2.5}$ prediction model at the second stage (CV $R^2$: 0.83–0.88) (Table S5). It was found that using AOD products interpolated by XG or RF (XG-AOD or RF-AOD) was beneficial to short-term PM$_{2.5}$ prediction accuracy (CV $R^2$: both 0.88), and more accurate than when using original AOD data (CV $R^2$: 0.85). We, therefore, further compared XG-AOD and RF-AOD in different regions (Figure 3). Generally, PM$_{2.5}$ prediction models using either AOD product performed similarly and obtained CV $R^2$ higher than 0.88 in most regions of China, except for the northwest (approximately 0.77–0.78). Using XG-AOD was more stable (lower SD or narrower 95% CI for CV $R^2$) in the northwest. This was similar to spatial CV and temporal CV (spatial (temporal) CV $R^2$ = 0.83(0.83) for RF vs. 0.82(0.82) for XG), while the PM$_{2.5}$ prediction model using the original AOD had a less stable performance, with a spatial CV $R^2$ of 0.65 and a temporal CV $R^2$ of 0.61 (Table S6).

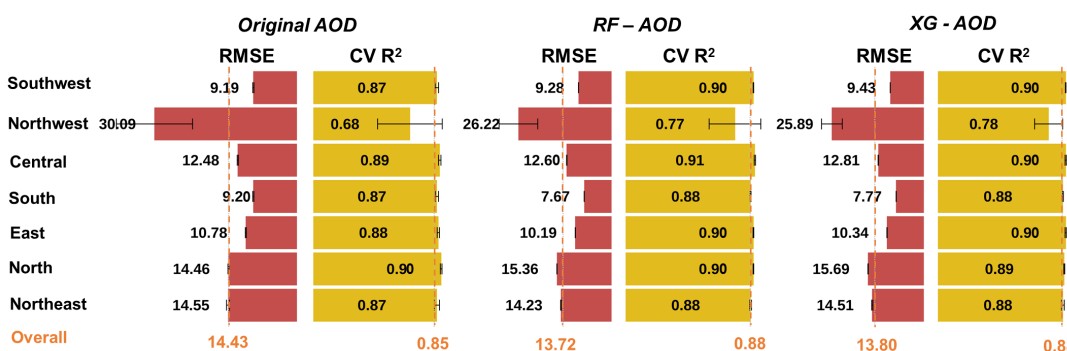

**Figure 3.** Cross-validation performance of PM$_{2.5}$ prediction model with AOD products interpolated by using random forest (RF) and extreme gradient boosting (XG). Orange dashed line: the overall performance; error bar: 95% CI.

### 3.4. Spatial and Temporal Cross-Validation

Due to the high demand of spatial extrapolation, a LOOCV$_{site}$ analysis was conducted (Figure 4). The results were similar while using two different interpolated AOD products at the PM$_{2.5}$-prediction stage. The model using RF-AOD performed slightly better than XG-AOD (median LOOCV $R^2$ (RMSE): 0.81(14.68 μg/m$^3$) vs. 0.78(15.07 μg/m$^3$), respectively). Both were significantly better than using original AOD (median LOOCV $R^2$ = 0.56, RMSE = 24.48 μg/m$^3$). The grids with higher accuracy (coverage rate and LOOCV $R^2$ > median level) were mainly located in the east of China, including Central, South, East, North, and Northeast China. For temporal CV, the RF or XG interpolation strategy performed significantly better than the original AOD, and they had similar temporal CV among different months or years (Figure S3). The warm season (April–September) generally performed more poorly than the cold season (November–March). The CV result was stable in different years.

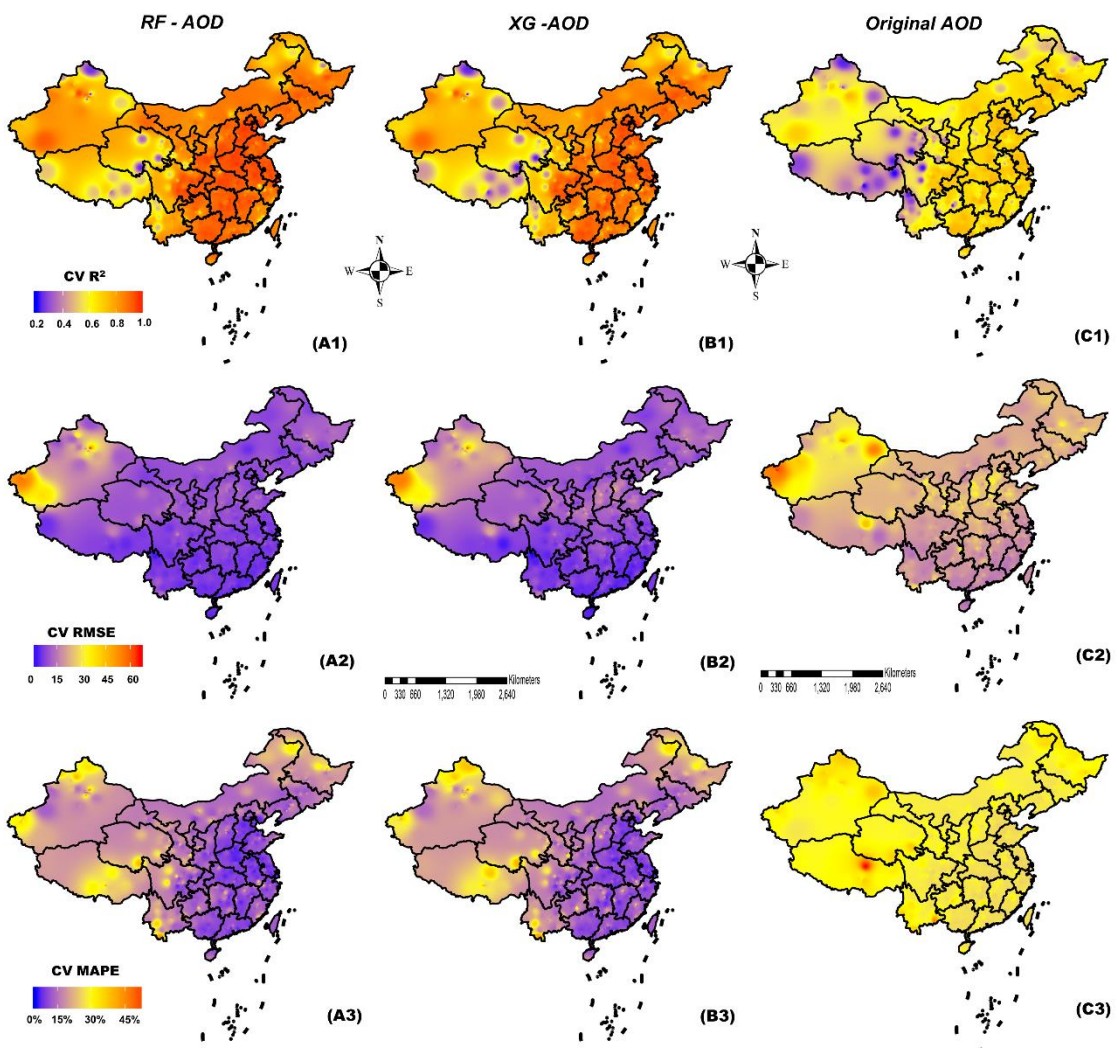

**Figure 4.** Leave-One-Out-Cross-Validation (LOOCV) performance (CV $R^2$, RMSE and MAPE) of daily PM$_{2.5}$-prediction models in China using RF-AOD (**A1**–**A3**), XG-AOD (**B1**–**B3**), and original AOD (**C1**–**C3**).

To better clarify the performance of the interpolation methods, we specifically illustrated the result under the worst condition during the study period, i.e., when the coverage of MAIAC AOD was the lowest (4.55% on 3 July 2018) (Figure 5). The coverage of AOD could reach 96.25% for both RF- and XG-interpolation (determined by the missing of external information), and median levels of

interpolated AOD changed from 0.18 to 0.53 (RF) and to 0.57 (XG). In the meantime, the coverage of predicted PM$_{2.5}$ increased from 4.12% to 95.46% (determined by the additional missing while considering lag effect), and LOOCV$_{site}$ R$^2$ of predicted PM$_{2.5}$ was 0.70 (original), 0.86 (RF), and 0.85 (XG). The median (IQR) levels of PM$_{2.5}$ (µg/m$^3$), estimated by MAIAC AOD, RF-AOD, and XG-AOD, were 36.34 (22.26), 27.52 (25.68), and 26.32 (27.42), respectively.

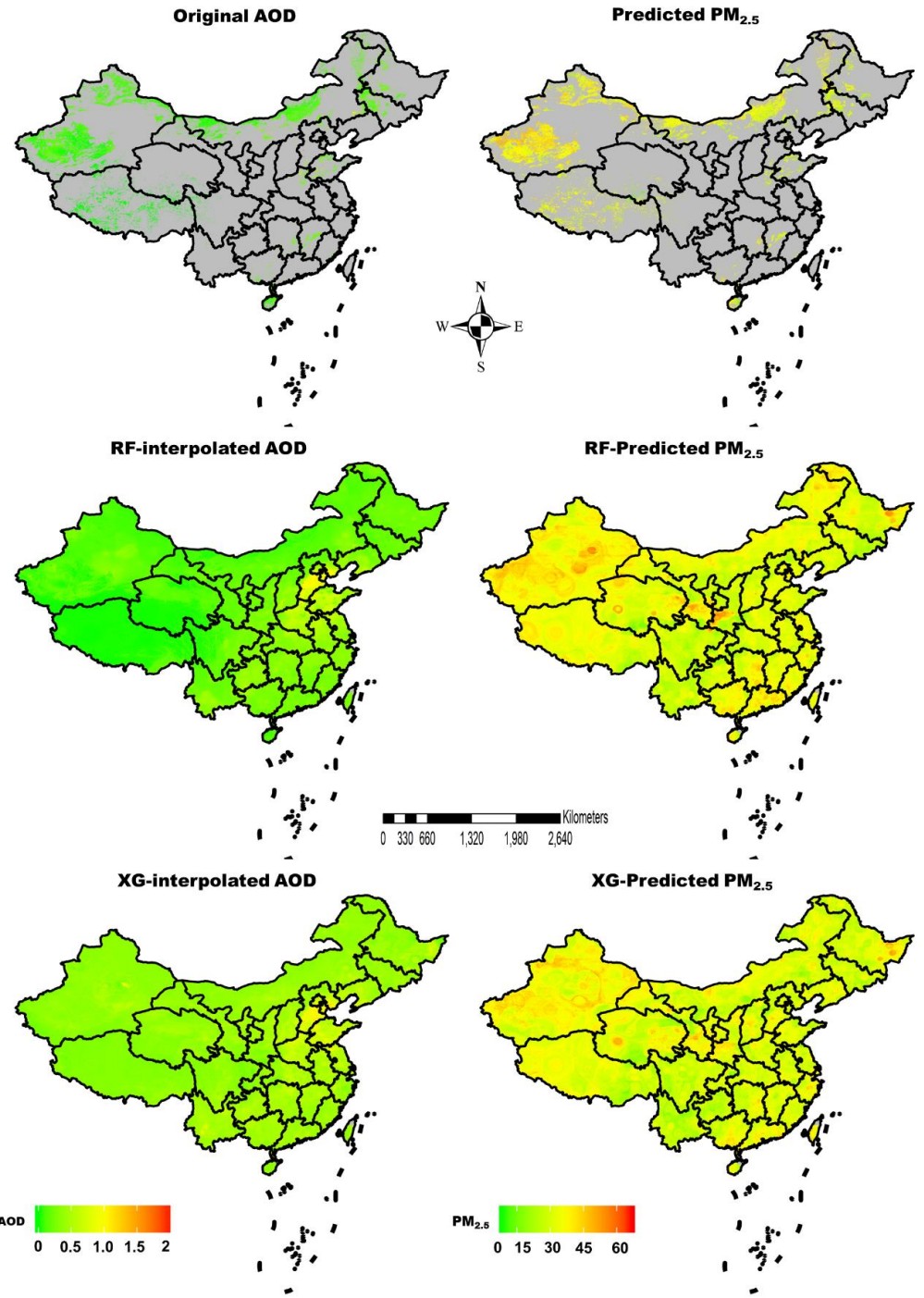

**Figure 5.** Levels of the observed and interpolated MAIAC AOD, and predicted PM$_{2.5}$ on 3 July 2018, with the lowest coverage of observed AOD.

## 4. Discussion

This study developed a series of strategies for estimating daily $PM_{2.5}$ exposure. This included two stages; the AOD-imputation stage and the $PM_{2.5}$ predicted stage. For the AOD-imputation stage, and to resolve the difficult aspects of our previous two-step interpolation such as relatively tedious steps and ignoring influences of external information [9], this study compared 12 different algorithms and found ML algorithms (with external information) generally achieved higher coverage than geostatistical algorithms (98.64% vs. 21.43–87.31%). The RF or XG interpolation among ML algorithms also guaranteed a higher interpolated quality (CV $R^2$ = 0.89 and 0.85) than other ML algorithms (CV $R^2$ = 0.49–0.72) or geostatistical algorithms (CV $R^2$ = 0.49–0.78). However, XG interpolation can better balance the computation time and performance. Compared with the model using original AOD, the $PM_{2.5}$ prediction model using XG-AOD not only guaranteed higher coverage (97.83% vs. 14.35%), but also a better prediction performance (CV $R^2$: 0.88 vs. 0.85) for the $PM_{2.5}$-prediction stage.

The AOD model has provided optimism for high-coverage estimations of $PM_{2.5}$ exposure to compensate for the very low coverage of existing ground air quality monitoring sites. It was found, however, that the AOD-missing rate was above 84% among the day-grid units in China during 2013–2018, and similar problems have been reported in different regions [14,39–41]. This raises debate regarding whether the short-term estimation of $PM_{2.5}$, based on retrieved AOD data, is more practical for further research rather than data from the sites. For example, in China, $PM_{2.5}$ predicted by MAIAC AOD still maintains a higher resolution (1 km) and a higher coverage (approximately 96,534,23 grids) than data from the sites (approximately 1605 sites), and the sites are more likely to be concentrated in eastern China, which is densely populated due to economic and social reasons [40]. A similar situation also exists in many developing countries and in high-income countries [42,43]. Missing observations of MAIAC AOD also tend to be concentrated in specific areas due to uncontrollable factors such as orbit patterns and cloudiness [9,10]. For example, in the lowest coverage day (3 July 2018) during the study period, severe convective weather and heavy rainfall in coastal and central areas of China may have led to such an absence. Furthermore, previous studies have suggested that a monsoon climate is associated with more frequently missed observations. The large seasonal variations in the observed aerosol data [17,44] due to more rainy or cloudy weather in the summer [45], means that regions with a monsoon climate, which are concentrated in densely populated areas, are more susceptible to the missing observation problem and therefore, appropriate AOD interpolation is a prerequisite before using the AOD model for higher-coverage assessments of $PM_{2.5}$ exposure.

Many attempts have previously been made to impute AOD-missing observations, but it remains unclear which is the most optimal. We compared the performances of 12 approaches, including those proposed here, and found ML algorithms using external information can fill more missing data than geostatistical interpolation (approximately 98.64% vs. 21.43–87.31%). This was consistent with previous assumptions [46–48], but the interpolation quality (CV $R^2$: 0.49–0.89) varied widely in different algorithms, with a similar range (CV $R^2$ approximately 0.34–0.85) reported in previous different studies [14–17,43,46–51]. The higher missing-filled efficiency of ML algorithms indicates that external information can provide more information at the AOD-imputation stage, because the geostatistical interpolation is over-dependent on spatial autocorrelation but ignores other influencing factors on the temporal dimension [21]. Not all ML algorithms, however, can obtain higher interpolation quality than geostatistical interpolation, and the varied interpolation quality suggests that the model structure, or the ability to capture complex spatio-temporal relationships rather than external information, plays a more important role in the accuracy of the AOD-filled process. Therefore, suitable ML algorithms were selected by 10-fold cross-validation. Most geostatistical interpolations could not provide a fixed model for the entire period, that is, they needed to be developed separately for each map slice at different time points, and so tended to take a longer running time. Among different interpolations, the RF and XG methods were the best two groups, with the highest CV $R^2$ (0.89 and 0.85, respectively), but the time consumption of the RF method was much longer due to parallel operation and different calculation

methods. Furthermore, the RF method consumed more memory during computation. It is necessary to tradeoff time consumption and interpolation quality if robust computing power is not available.

It was also found that the average AOD levels always rose after interpolation, irrespective of the interpolating method used. This is consistent with the phenomenon that missing more frequently occurs at high MODIS-AOD levels [52,53]. The aerosol hygroscopic growth during missing days results from increased humidity from cloud or humid airflows [54,55]. Another potential reason is that the surface reflectivity, cloudiness, or snow is always associated with over-exposure values [9,10], so it tends to be eliminated during the preprocess of satellite sensors or systems. It was also found that $PM_{2.5}$ levels on AOD-missing days were lower than those on non-missing days (36.0 µg/$m^3$ vs. 39.0 µg/$m^3$), and this was partially due to shorter particle suspension time with heavy rainfall, convective weather or snow [56], and higher missing rates in the warm season (April–September), with relatively lower $PM_{2.5}$ levels (Figure 2). This suggests that the relationship between AOD and $PM_{2.5}$ after AOD interpolation, becomes more complicated than a simple linear relationship [9]. This could explain why AOD interpolation can lead to weaker linear correlations (0.43 to 0.31) (Table S4) but higher accuracy in our $PM_{2.5}$ prediction model, which can capture the complicated spatio-temporal variations (0.85 to 0.88) (Figure 3).

At the $PM_{2.5}$ predicted stage, further discussion is required regarding how many benefits can be obtained from different interpolated AOD products. Compared with non-AOD model, the imputed AOD always improves the accuracy of $PM_{2.5}$ estimation (CV $R^2$ $_{PM2.5 estimation}$ = 0.77 vs. 0.83~0.88) (Table S5), especially for XG and RF interpolations which had higher interpolating quality (CV $R^2$ $_{AOD imputation}$ > 0.8). Furthermore, compared with RF-AOD and XG-AOD, the satellite $PM_{2.5}$ model using original AOD performed relatively poorly in spatial and temporal CVs (spatial (temporal) CV $R^2$ = 0.83(0.83) vs. 0.82(0.82) vs. 0.65(0.61)). The lower LOOCV $R^2$ (=0.56) when only using original AOD further proved its poorer spatial application to all mainland China because different missing rates in different grids can increase the heterogeneity (or uncertainty) of predictions, and further weaken spatial extrapolation to other grids. Furthermore, incomplete time series of AOD produced for each grid also makes it more difficult to obtain an accurate prediction in any out-of-sample time. Compared with the XG method, however, the higher accuracy of RF methods at the AOD-imputation stage did not bring more benefits to the $PM_{2.5}$-prediction stage. It indicated that the missing rate of the AOD product, compared with the accuracy of the interpolated AOD product, is the key to spatial and temporal extrapolation of the $PM_{2.5}$-prediction stage.

Some limitations were present in this study. First, the complex relationships between AOD-missing effect and $PM_{2.5}$ may cause annual and regional differences, which need to be further analyzed in more countries for a longer research period. Second, the AOD miss-filling approaches taken here are based on using known predictors which are easily accessed and are frequently recorded in most areas and therefore, some other chemical or physical features such as chemical composition in aerosol and sun radiation were not considered in this study. Finally, the LOOCV result for all of China was interpolated by UK, and therefore, some unavoidable bias due to the UK technique may be present. This result provides an approximate assessment only for spatial dimensions.

## 5. Conclusions

This study proposed a XGBoost method to impute missing data of AOD based on some external predictors. By comparing a variety of different imputation methods, XGBoost is confirmed to be a less consuming-time choice, with almost full coverage, good missing-imputation quality, and consequently, accurate prediction of $PM_{2.5}$. In terms of practicality, our study provides some guidance, strategies, and a tempo-spatially continuous $PM_{2.5}$ dataset for future short-term health impact assessments in epidemiological studies of air pollution.

**Supplementary Materials:** The following are available online at http://www.mdpi.com/2072-4292/12/18/3008/s1, Figure S1. Geographical distribution of $PM_{2.5}$ monitoring sites (A) and meteorological stations (B) in mainland China in 2015. Figure S2. Geographical locations of regions and provinces in China. Figure S3. The temporal

CV performance (Proportion of sites with different CV R$^2$) of daily Satellite-PM$_{2.5}$ model in China with RF-AOD, XG-AOD and Original AOD. Table S1. The optimal parameters selected in AOD-filled Stage and PM$_{2.5}$ Predicted Stage. Table S2. Optimal combination of variables selected in different regions of China. Table S3. The spatial and temporal CV of Geostatistical and ML interpolation for MAIAC AOD product. The comparison of daily Satellite-PM$_{2.5}$ model with AOD products interpolated by using interpolating methods (with CV R$^2$ for interpolation > = 0.7). Table S6. The spatial and temporal CV of using different interpolated-AOD product in PM$_{2.5}$ Predicted Stage.

**Author Contributions:** Conceptualization, Z.-Y.C., J-Q.J., R.Z., T.-H.Z., J-J.C., J.Y., C.-Q.O. and Y.G.; Data curation, Z.-Y.C., J-Q.J., J-J.C. and J.Y.; Formal analysis, Z.-Y.C.; Funding acquisition, C.-Q.O. and Y.G.; Methodology, Z.-Y.C., R.Z. and C.-Q.O.; Project administration, Z.-Y.C., C.-Q.O. and Y.G.; Software, Z.-Y.C.; Supervision, C.-Q.O. and Y.G.; Validation, Z.-Y.C., J-Q.J., R.Z. and T.-H.Z.; Writing—original draft, Z.-Y.C.; Writing—review and editing, Z.-Y.C., C.-Q.O. and Y.G. All authors have read and agreed to the published version of the manuscript.

**Funding:** This research was funded by National Nature Science Foundation of China [81573249], Nature Science Foundation of Guangdong Province [2016A030313530] and Australian National Health and Medical Research Council [APP1107107]. And The APC was funded by National Nature Science Foundation of China [81573249] and Nature Science Foundation of Guangdong Province [2016A030313530].

**Acknowledgments:** This study was supported by National Nature Science Foundation of China [81573249], Nature Science Foundation of Guangdong Province [2016A030313530]. YG was supported by Career Development Fellowship of Australian National Health and Medical Research Council [APP1107107].

**Conflicts of Interest:** The authors declare no conflict of interest.

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
