# Peer review of "Comparison of Different Missing-Imputation Methods for MAIAC (Multiangle Implementation of Atmospheric Correction) AOD in Estimating Daily PM2.5 Levels"

_remotesensing, doi:10.3390/rs12183008_

Round 1
Reviewer 1 Report
I noticed that the authors have improved the manuscript. However, there is sill insufficient background. For example, the two-stage method, which was said in Abstract to be the aimed method to be developed in this paper, was not introduced at all in Introduction but showed up in the last paragraph. The authors need to introduce more background so that the audience have an idea why previous methods are not good enough and how the new method can improve it.
The methods and results seemed to be more improved, though I am still confused how the predictors were selected. Are they randomly selected or based on some criteria? The authors should at least give some discussion on these.
The conclusions seemed that the main purpose of this paper is to establish a framework for comparison of existing methods, but not developing a new method, as stated in abstract. Please try to be consistent.
Author Response
Reviewer 1
I noticed that the authors have improved the manuscript. However, there is still insufficient background. For example, the two-stage method, which was said in Abstract to be the aimed method to be developed in this paper, was not introduced at all in Introduction but showed up in the last paragraph. The authors need to introduce more background so that the audience have an idea why previous methods are not good enough and how the new method can improve it.
[RESPONSE] Thanks for the kindly comments. As suggested, we have clearly pointed out the weakness of previous methods and the significance of the new method we will propose in the revised manuscript (32-33, 91-93, 110-116, 467-469). This study developed a two-stage approach (AOD-imputation stage and PM2.5-prediction stage). Actually, Our main purpose is to improve the first stage (AOD-imputation stage) by comparing various imputation methods, while at the second stage we applied the previous method we developed[1] to analyze how imputation method for AOD affect the final prediction of PM2.5 . Previous AOD-imputation methods were over-dependent on geostatistical interpolation, and some external information, such as meteorological data and cloud fraction, were not taken into account, leading to relatively low coverage and imputation quality of AOD which subsequently limit the accuracy of PM2.5 prediction where AOD is an important input variable. We attempted to apply machine learning methods to capture complex non-linear relationships with external predictors. A variety of machine learning algorithms and geostatistical approaches were compared to figure out an optimal method for AOD imputation and final prediction of PM2.5 (95-98, 100-113).
The methods and results seemed to be more improved, though I am still confused how the predictors were selected. Are they randomly selected or based on some criteria? The authors should at least give some discussion on these.
[RESPONSE] The predictors were selected according to the following criteria which were described in details in the revised manuscript “At the AOD-imputation stage, the external predictor was selected by multiple linear models if it was significantly associated with AOD (P< 0.05) and had no multi-collinearity problem (low variance inflation factor (VIF) <5).” (191-193)". “At the PM2.5-prediction stage, non-lagged variable (e.g. land-related data) and potential lagged variable (e.g. meteorological variables) were selected by the NELRM [2] if it was significantly associated with PM2.5 (P< 0.05) and had no multi-collinearity problem (VIF<5).” (220-223).
The conclusions seemed that the main purpose of this paper is to establish a framework for comparison of existing methods, but not developing a new method, as stated in abstract. Please try to be consistent.
[RESPONSE] Thanks for your reminder. The main purpose is to develop a new method. We had made them consistent. (32-33, 467-469)
Reference
- Chen, Z.-Y.; Zhang, T.-H.; Zhang, R.; Zhu, Z.-M.; Yang, J.; Chen, P.-Y.; Ou, C.-Q.; Guo, Y. Extreme gradient boosting model to estimate PM2. 5 concentrations with missing-filled satellite data in China. Atmos. Environ. 2019, doi:10.1016/j.atmosenv.2019.01.027.
- Chen, Z.; Zhang, T.; Zhang, R.; Zhu, Z.; Ou, C.; Guo, Y. Estimating PM2.5 concentrations based on non-linear exposure-lag-response associations with aerosol optical depth and meteorological measures. Atmos. Environ. 2018, 173, 30–37, doi:10.1016/j.atmosenv.2017.10.055.
Reviewer 2 Report
The manuscript has been improved and includes more details for methods and discussion. The authors also separate the approach into AOD-imputation stage and PM2.5-prediction stage, which is easier to read. Overall, this manuscript compares geostatistic interpolation and machine learning algorithms to estimate AOD and PM2.5, and it will be interesting for readers.
Author Response
Reviewer 2
The manuscript has been improved and includes more details for methods and discussion. The authors also separate the approach into AOD-imputation stage and PM2.5-prediction stage, which is easier to read. Overall, this manuscript compares geostatistic interpolation and machine learning algorithms to estimate AOD and PM2.5, and it will be interesting for readers.
[RESPONSE] We appreciate this encouraging comment.
Reviewer 3 Report
The paper under review “Comparison of different missing imputation methods for MAIAC 1 AOD in estimating daily PM2.5 level” provides a comprehensive study on the potential effects of missing Satellite derived (MAIAC) AOD data in the final performance of Model Based daily PM25 estimation where AOD is an important input variable. In particular, the authors present a large number of Geostatistical and Machine Learning approaches to filling in the missing (cloud masked) AOD data and proceed to explore how the PM25 estimation model performs through multiple cross-validation exercises.
The paper is generally well written and organized which is critical since the paper is very dense in describing the relative performance metrics trying to balance between different performance metrics such as linear correlation, RMSE, Data Processing time, coverage %, etc. As expected, no one approach is clearly better than the rest and the authors do a good job of describing in words and tables the relative benefits/weaknesses
There are however, some minor issues that I think should be cleared up to improve the readability of the paper
- It is not clear to me what is meant by Temporal Cross-Validation (CV) The authors provide reference 14 but that does not help. in describing this concept. I think it would be very helpful if the concept and methodology is defined and explained and the actual mechanics described. It seems this can be done in Section 2.2.3 Validation stage which should be expanded so all the Validation Methodologies are self-contained in one subsection
- The actual multidimensional training/validation dataset sizes are often described as parenthetical numbers but it is not so easy for the reader to keep track of what the numbers mean Ex on lines 198-200. “It should be noted that the sample size of MAIAC AOD covering China from 2013–2018 was too large (approximately 96,534, 23× 365× 6)”. The actual meaning of some of the dimension sizes are clear but some are not so it would be helpful for the authors to define them
- I believe when doing the final evaluation, the CV performance is based on the surface PM25 network only and it does make sense that this is the highest priority to evaluate since this is the output product. However, it seems that some evaluation of the intermediate “imputated” AOD values themselves should be evaluated by some CV approach to see if the intermediate performance may explain some of the output PM25 model performance
- The authors refer to multiple plots not in the paper using the enumeration Figure A.1 etc. Usually, this is referring to an appendix but here, it is referring to the supplemental information paper. The Author should remind the reader of this.
Author Response
Reviewer 3
The paper under review “Comparison of different missing imputation methods for MAIAC 1 AOD in estimating daily PM2.5 level” provides a comprehensive study on the potential effects of missing Satellite derived (MAIAC) AOD data in the final performance of Model Based daily PM25 estimation where AOD is an important input variable. In particular, the authors present a large number of Geostatistical and Machine Learning approaches to filling in the missing (cloud masked) AOD data and proceed to explore how the PM25 estimation model performs through multiple cross-validation exercises.
The paper is generally well written and organized which is critical since the paper is very dense in describing the relative performance metrics trying to balance between different performance metrics such as linear correlation, RMSE, Data Processing time, coverage % etc. As expected, no one approach is clearly better than the rest and the authors do a good job of describing in words and tables the relative benefits/weaknesses
[RESPONSE] We appreciate reviewer’s encouraging comment.
There are, however, some minor issues that I think should be cleared up to improve the readability of the paper
It is not clear to me what is meant by Temporal Cross-Validation (CV) The authors provide reference 14 but that does not help in describing this concept. I think it would be very helpful if the concept and methodology is defined and explained and the actual mechanics described. It seems this can be done in Section 2.2.3 Validation stage which should be expanded so all the Validation Methodologies are self-contained in one subsection
[RESPONSE] Done. We had added the detailed description in the revised manuscript (240-252).
The actual multidimensional training/validation dataset sizes are often described as parenthetical numbers but it is not so easy for the reader to keep track of what the numbers mean Ex on lines 198-200. “It should be noted that the sample size of MAIAC AOD covering China from 2013–2018 was too large (approximately 96,534, 23× 365× 6)”. The actual meaning of some of the dimension sizes are clear but some are not so it would be helpful for the authors to define them
[RESPONSE]Done. We had revised it as suggested for example, ” (approximately 96,534,23 (numbers of grids)365 (days)6 (years)))” (201).
I believe when doing the final evaluation, the CV performance is based on the surface PM25 network only and it does make sense that this is the highest priority to evaluate since this is the output product. However, it seems that some evaluation of the intermediate “imputated” AOD values themselves should be evaluated by some CV approach to see if the intermediate performance may explain some of the output PM25 model performance
[RESPONSE]Thanks for your suggestions. The cross-validation results of the “imputated” AOD values themselves by different interpolation are shown in Table 2. And the CV performance of PM2.5 estimates using different “imputated” AOD (those with CV R2 for interpolation>=0.7) are shown in Table S.5. Indeed, And we also added the performance of non-AOD model in Table S.5 and described as “Compared with non-AOD model, the imputed AOD always improves the accuracy of PM2.5 estimation (CV R2 PM2.5 estimation = 0.83~0.88 VS 0.77) (Table S.5), especially for XG and RF interpolations which had higher interpolating quality (CV R2 AOD imputation >0.8).”(440-443)
The authors refer to multiple plots not in the paper using the enumeration Figure A.1 etc. Usually this is referring to an appendix but here, it is referring to the supplemental information paper. The Author should remind the reader of this.
[RESPONSE] Done.
Round 2
Reviewer 1 Report
The authors addressed my previous questions. I have no further questions.
This manuscript is a resubmission of an earlier submission. The following is a list of the peer review reports and author responses from that submission.
Round 1
Reviewer 1 Report
The present study aims to develop a less time consuming, higher coverage and more accurate interpolation method for high missing MAIAC AOD data in china during 2013-2018. The study aims also to evaluate whether the interpolation method affect PM2.5 concentrations.
At the beginning, the manuscript looks scientifically significant. However, after checking the literature, I would suggest to reject the manuscript. The findings do not add any new contribution to the journal and the results are redundant. Similar study with almost the same authors was conducted using the same model NELRM in Chen et al. (2019) but using a different dataset. Both studies prove that XG boost has better performance to predict PM2.5. Extending the study period (2013-2018) using the same methodology as used in Chen et al. (2019) is not enough to be published, even though the authors present a comparison of the different interpolation methods in the manuscript.
In addition, the manuscript in its current state is not ready for peer review. The English in this paper needs major improvement and the references are not all compiled in the right form of MDPI references style (lines 69,122,303,345).
Major comments: The modelling framework was not presented, so it was difficult for me to follow and to understand the methodology part. The results are poorly interpreted. The discussion is too long as the result part was not efficiently analysed. However, a discussion about the uncertainty of the predicted PM2.5 was missing, in other terms the limitation and prospects of the methodology used.
Please use the right reference to cite MAIAC data (Lyapustin et al, 2011; Lyapustin et al, 2018).
Zhao-Yue Chen, Tian-Hao Zhang, Rong Zhang, Zhong-Min Zhu, Jun Yang, Ping-Yan Chen, Chun-Quan Ou, Yuming Guo, Extreme gradient boosting model to estimate PM2.5 concentrations with missing-filled satellite data in China,Atmospheric Environment, Volume 202,2019,https://doi.org/10.1016/j.atmosenv.2019.01.027.
Lyapustin, A., Wang, Y., Laszlo, I., Kahn, R., Korkin, S., Remer, L., Levy, R., and Reid, J.: Multiangle implementation of atmospheric correction (MAIAC): 2. Aerosol algorithm, J. Geophys. Res.-Atmos., 116, D03211, https://doi.org/10.1029/2010JD014986, 2011.
Lyapustin, A.,Wang, Y., Korkin, S., and Huang, D.: MODIS Collection 6 MAIAC algorithm, Atmos. Meas. Tech., 11, 5741–5765,https://doi.org/10.5194/amt-11-5741-2018, 2018.
Reviewer 2 Report
- The authors should explain more why they have to use or compare MAIAC AOD and Satellite-PM2.5 model. What’s difference between them? If MAIAC AOD interpolation could have high R2, why do you need Satellite-PM2.5 model? I felt confused as they all include meteorological data and have high R2. Which one is better?
- The authors suggested that XG method is better as it has high R2 but less computing time than RF method. However, the authors just only introduced RF processing in the method. If XG is better, the authors should introduce XG in the method as well.
- In L343-349, “we found the AOD levels on average always rise after interpolation, no matter what interpolating methods were used.” The authors’ explanation is “It is probably because the over-exposure or difficultly-observed values, affected by some factors associated with AOD vertical profiles like cloudiness or snow, could be eliminated during the collecting data process of satellite sensors and further retrieving data.” However, the authors don’t have persuaded evidences to support their interpretations. How do you process the error or noise? Is it possible to cause this increase? The authors should provide more evidences to support this explanation.
- The authors used 12 interpolation method, but didn’t introduce them in the method except RF method and have less discussion about other methods in the results and discussion. That’s quite weird. Maybe they can provide a summary table to introduce these methods in the method. Otherwise, it is meaningless to include these methods in the manuscript.
Reviewer 3 Report
The authors tried to compare 12 methods along with their proposed one for interpolating satellite PM2.5 data to fill missing values. The manuscript is hard to read. The methods and results were poorly presented. It needs significant improvement of writing before further consideration for publication. One example out of many: in Equation 2, which parameters are predictors as referenced in the context?